# Clinical Features, Inpatient Trajectories and Frailty in Older Inpatients with COVID-19: A Retrospective Observational Study

**DOI:** 10.3390/geriatrics6010011

**Published:** 2021-02-01

**Authors:** Christopher N. Osuafor, Catriona Davidson, Alistair J. Mackett, Marie Goujon, Lelane Van Der Poel, Vince Taylor, Jacobus Preller, Robert J. B. Goudie, Victoria L. Keevil

**Affiliations:** 1Department of Medicine for the Elderly, Addenbrooke’s Hospital, Cambridge University Hospitals NHS Foundation Trust, Hills Road, Cambridge CB2 0QQ, UK; catriona.davidson@addenbrookes.nhs.uk (C.D.); alistair.mackett@addenbrookes.nhs.uk (A.J.M.); marie.goujon@addenbrookes.nhs.uk (M.G.); lelane.vanderpoel@addenbrookes.nhs.uk (L.V.D.P.); victoria.keevil@addenbrookes.nhs.uk (V.L.K.); 2Department of Clinical Neurosciences, University of Cambridge, Cambridge CB2 0QQ, UK; 3Cancer Research UK, Cambridge University Hospitals NHS Foundation Trust, Cambridge CB2 0QQ, UK; vt217@medschl.cam.ac.uk; 4Department of Acute Internal Medicine and Intensive Care, Cambridge University Hospitals NHS Foundation Trust, Cambridge CB2 0QQ, UK; kobus@preller.me.uk; 5Medical Research Council Biostatistics Unit, University of Cambridge, Cambridge CB2 0SR, UK; robert.goudie@mrc-bsu.cam.ac.uk; 6Department of Medicine, University of Cambridge, Cambridge CB2 0QQ, UK; 7Department of Public Health and Primary Care, University of Cambridge, Cambridge CB2 0SR, UK

**Keywords:** atypical presentation, COVID-19, frailty, mortality, older patients

## Abstract

Introduction: We describe the clinical features and inpatient trajectories of older adults hospitalized with COVID-19 and explore relationships with frailty. Methods: This retrospective observational study included older adults admitted as an emergency to a University Hospital who were diagnosed with COVID-19. Patient characteristics and hospital outcomes, primarily inpatient death or death within 14 days of discharge, were described for the whole cohort and by frailty status. Associations with mortality were further evaluated using Cox Proportional Hazards Regression (Hazard Ratio (HR), 95% Confidence Interval). Results: 214 patients (94 women) were included of whom 142 (66.4%) were frail with a median Clinical Frailty Scale (CFS) score of 6. Frail compared to nonfrail patients were more likely to present with atypical symptoms including new or worsening confusion (45.1% vs. 20.8%, *p* < 0.001) and were more likely to die (66% vs. 16%, *p* = 0.001). Older age, being male, presenting with high illness acuity and high frailty were independent predictors of death and a dose–response association between frailty and mortality was observed (CFS 1–4: reference; CFS 5–6: HR 1.78, 95% CI 0.90, 3.53; CFS 7–8: HR 2.57, 95% CI 1.26, 5.24). Conclusions: Clinicians should have a low threshold for testing for COVID-19 in older and frail patients during periods of community viral transmission, and diagnosis should prompt early advanced care planning.

## 1. Introduction

In December 2019, doctors in China described a cluster of viral pneumonia cases secondary to a novel Coronavirus named severe acute respiratory syndrome coronavirus 2 (SARS-CoV-2) [1]. The World Health Organization named this emergent disease coronavirus disease 19, or “COVID-19”, and declared a pandemic on 11 March 2020 [2]. One of the main challenges of COVID-19 is the wide spectrum of clinical presentations, ranging from asymptomatic cases to mild upper-respiratory-tract illness with myalgia and fatigue [3,4], severe viral pneumonia [1,3,4,5], gastrointestinal symptoms [5,6], neurological symptoms [7,8], cardiovascular symptoms [9,10] and other symptoms such as loss of smell and taste [11,12]. Atypical presentations [5] such as delirium [13,14], drowsiness [15], syncope [16] and fall [17] have also been described in older patients (aged ≥ 65 years), and our experience with other respiratory viral illnesses is that they often present without the typical fever and respiratory features [18,19]. Indeed, many illnesses in older adults present atypically in the form of a Geriatric Syndrome such as delirium, falls or reduced physical functional ability [20,21]. Understanding the spectrum of disease presentation is particularly important in COVID-19, given the necessity for prompt isolation of cases in order to prevent nosocomial spread and unnecessary exposure of health and social care staff.

Older adults are especially vulnerable to COVID-19 and experience high morbidity and mortality as a result of infection [22]. Age is independently linked with mortality [22,23], but age alone does not adequately capture the robustness of older adults who are a heterogeneous group. Some live well into old age and do not experience the “slowing up” and vulnerabilities associated with the onset of frailty whilst others do [24,25]. Age alone is also not adequate to determine the likely effectiveness of treatment options and assess prognosis, as well as to make decisions regarding healthcare resource allocation [26].

Over the past few decades, there has been an explosion of research into frailty and its operationalisation in the clinical setting. Assessment of frailty in older adults is now an important part of Comprehensive Geriatric Assessment and it is necessary to understand the relationships between frailty and both the clinical presentations and outcomes of COVID-19. One study in patients over 80 years old showed mortality from COVID-19 was significantly higher in patients with frailty [23], another showed frailty was weakly associated with higher risk of mortality [27] while a third study found that frailty is not a good discriminator of prognosis in COVID-19 [28].

In this retrospective observational study, we aim to describe the range of clinical presentations of older patients (≥65 years) admitted as an emergency to a University Hospital and diagnosed with COVID-19. Information retrieved from our electronic health record system will provide a detailed description of the presenting symptoms, radiological findings and laboratory characteristics of this patient group, as a whole and by frailty status. We will also describe the clinical trajectory of patients from admission to discharge in terms of delirium, functional status and hospital outcomes, particularly exploring associations between frailty and mortality in this cohort. This will add to our understanding of how this emergent infectious disease affects older adults.

## 2. Materials and Methods

### 2.1. Study Design and Setting

This is a retrospective observational study conducted in a tertiary National Health Service (NHS) University Hospital in England.

### 2.2. Sample

Any older adult (≥65 years old) admitted as an emergency from 1 March 2020 to 15 May 2020 and diagnosed with COVID-19 infection was included. Diagnosis was confirmed either by clinical suspicion of probable COVID-19 infection and/or positive viral Polymerase Chain Reaction (PCR) for SARS-COV-2 viral RNA on nasopharyngeal swab.

### 2.3. Patient Characteristics

The following information was retrieved by the clinical informatics team from the Electronic Health Record (EHR): age, sex, ethnicity, body mass index (BMI, Kg/m^2^), Clinical Frailty Scale (CFS) score [25], number of ward moves (excluding the Emergency Department or equivalent acute assessment area), PCR for SARS-COV-2 viral RNA results and laboratory blood tests on admission. Laboratory values included C-reactive protein (CRP; mg/L), total white blood cell (WBC), lymphocyte and neutrophil counts (10^9^/L), interleukin-6 (IL-6; pg/mL), creatinine (µmol/L), high sensitivity troponin I (HsTNI; ng/L) and D-dimer (ng/mL). The maximum National Early Warning Score 2 (NEWS2) [29] and associated vital signs (heart rate, systolic blood pressure, respiratory rate and temperature) recorded in the first 4 h of admission were also retrieved. The CFS is a 9-point measure of the overall level of fitness or frailty of an older adult. It is based on clinical judgement of the patient’s status two weeks before the onset of the acute illness leading to hospital admission. Higher scores indicate increasing frailty. The NEWS2 is an aggregate score based on the deviation of bedside vital signs from a normal physiological range. Higher scores indicate higher illness acuity.

Further clinical information was manually extracted from the EHR using a standardised data collection tool. Presenting symptoms (history of fever, cough, breathlessness, fatigue, myalgia, new or worsening confusion, fall, nausea and/or vomiting, diarrhoea, abdominal pain, loss of taste and/or smell, or nonspecific illness), number of regular admission medications, multimorbidity (<2, 2 or >2 long term conditions), known dementia, admission from a care home (residential versus dual registered/nursing homes), functional status on admission and discharge (independently mobile, mobile using a stick, mobile using a frame or immobile), admission chest X-ray (CXR) report [30] and evidence of hospital-acquired disease (defined as diagnosis of COVID-19 infection >14 days after hospital admission or if this was the diagnosis recorded by the treating team). Finally, whether the patient developed new or worsening confusion or was diagnosed with delirium during the admission episode was documented, after searching the EHR using the terms “confusion” and “delirium”. Patients were also assigned a CFS score by the researcher if this had not been recorded by the treating team in the EHR (102 (48%) patients). All CFS scores were completed using information recorded in the EHR on the patient’s admission to hospital and verified by a consultant or specialist registrar in Geriatric Medicine, experienced in the assessment of frailty using the CFS. CFS scores derived from manual review of inpatient records show good agreement with scores assigned after face to face assessment, with an estimated Kappa of 0.84 for inter-rater reliability [31]. All CFS scores were assigned as soon as possible after admission but it was not possible to blind researchers to the outcome if patients had already died, as users are automatically notified of this when the electronic patient record is accessed. Patients assigned a CFS score by the research team were younger than other patients (median (interquartile range) ages: 76 years, (71,83) versus 84 years, (77,89); *p* < 0.001) but there was no significant difference in sex (% women: 39.1% versus 47.3%, *p* = 0.27). The age difference reflects a policy at our centre, predating the pandemic, which only mandates frailty scoring for patients aged 75 years and over. Therefore, many patients aged 65–74 years old included in our cohort did not have a CFS score assigned by their treating team.

For patients identified with hospital-acquired COVID-19 disease, laboratory test results nearest to the first positive PCR for SARS-COV-2 viral RNA (within 12 h), the maximum NEWS2 score, vital signs nearest to the first positive PCR for SARS-COV-2 viral RNA (within 4 h) and presenting features prompting the treating team to consider a diagnosis of COVID-19 (including nearest CXR report) were used instead of results at hospital admission. All laboratory tests and vital signs were extracted by the clinical informatics team and presenting features were manually extracted from the EHR by the research team.

### 2.4. Patient Outcomes

Death during the inpatient stay or within 14 days of discharge was the primary outcome and was calculated using the dates of admission, discharge and death. The maximum follow-up time was 45 days for each patient, either 45 days from the date of admission or from the first positive swab in the case of hospital-acquired disease. It was important to include deaths in the two weeks following discharge, since local preparations for the COVID-19 pandemic included facilitation of early discharge to community palliative care settings.

Other hospital outcomes reported were prolonged length of stay (LOS; defined as a stay of ≥10 days), delayed discharge (defined as a stay >24 h after the last recorded “clinically fit date”, the date set by the treating team indicating fitness to leave hospital), 30-day readmission and new institutionalization (defined as a new admission to a care home). All hospital and out-of-hospital outcomes were provided by the clinical informatics team and available from the EHR.

### 2.5. Statistical Analysis

Descriptive characteristics were expressed as medians (interquartile range, IQR) and percentages (frequency, % (n)). The characteristics of the cohort were described as a whole and by frailty status. Frail (CFS ≥ 5) and nonfrail (CFS ≤ 4) patients were compared using Kruskal Wallis and Chi squared tests, and Venn diagrams were used to illustrate key differences. Cox proportional hazards regression further explored associations between frailty and mortality. Frailty was categorised as: CFS 1–4 “Nonfrail”, CFS 5–6 “Mild to Moderate Frailty” and CFS 7–8 “Severe to Very Severe Frailty”. Kaplan–Meier curves were explored to ensure that there was no violation of the proportional hazards assumption. Univariable and multivariable associations were explored after adjustment for age, sex, illness acuity and multimorbidity (defined as the coexistence of 2 or more long-term conditions). All analyses were performed using STATA (version 12).

## 3. Results

### 3.1. Patient Characteristics

There were 215 older inpatients eligible for this study and only 1 patient, who was terminally ill prior to developing COVID-19 and died on arrival to hospital, was excluded. A total of 213 (99.5%) patients had a PCR confirmed diagnosis while only 1 patient had a clinical diagnosis of COVID-19. Mean age was 80.3 years (range 65 to 103), 94 (43%) patients were women and 178 (83.2%) patients were White (Black: 1 [0.5%], Asian: 3 [1.4%]; other: 1 [0.5%], missing ethnicity or not stated: 31 [14.5%]). Table 1 further details the characteristics of the cohort by level of frailty.

### 3.2. Relationship between Clinical Characteristics and Frailty

Frail patients were significantly older and more likely to be women with lower BMI. They were also more likely to have dementia, multimorbidity and polypharmacy, and to be admitted from a care home. In terms of presenting symptoms, frail patients were less likely to complain of fever, cough, myalgia, fatigue, gastrointestinal symptoms or loss of taste and smell on presentation than nonfrail patients, but were more likely to have new or worsening confusion, an acute fall, or nonspecific illness as part of their presenting features. These latter three features are common acute geriatric illness presentations, and the differences in their prevalence between frail and nonfrail patients is striking (Figure 1).

There was no difference between frail and nonfrail patients in terms of their illness acuity on presentation of COVID-19 infection. Laboratory markers were also similar, with the exception of CRP and IL-6, which were higher in nonfrail patients (Table 1), and HsTNI, which was higher in frail patients. In terms of radiologic features, frail patients were more likely to have a normal CXR at the time of COVID-19 presentation compared to nonfrail patients. Similar proportions of frail and nonfrail patients acquired COVID-19 in hospital (Table 1).

### 3.3. Relationship between Inpatient Trajectory and Frailty

In terms of inpatient trajectory, frail patients were much less likely to be cared for in a high-care ward than nonfrail older adults but were much more likely to experience delirium (Table 2). There were 57.0% of frail patients who experienced new or worsening confusion, or received a diagnosis of delirium, during the admission episode compared to 29.2% of nonfrail patients. Frail patients were also more likely to be immobile on discharge and experience delayed discharge, remaining in hospital after their clinically fit date. There were also nonsignificant trends for longer lengths of inpatient stay, higher 30-day readmission and new institutionalisation amongst frail patients compared to nonfrail patients (Table 2).

### 3.4. Relationship between Mortality and Frailty

Patients who were frail were more likely to die during the inpatient episode or in the immediate two weeks following discharge (Table 2). A dose–response association was observed between higher frailty categories and higher mortality (Figure 2).

Associations between frailty and mortality were further explored using Cox (Proportional Hazards) regression (Table 3). The dose–response association between higher frailty and higher mortality persisted after adjustment for age, sex, illness acuity and multimorbidity.

## 4. Discussion

We have described the clinical features and inpatient trajectory of older inpatients with clinically confirmed COVID-19 admitted as an emergency to an NHS University Hospital in England. Our findings confirm the range of symptoms other than fever, cough and breathlessness that are common presenting features of COVID-19 infection in this population group. Furthermore, we found frail patients were more likely to present with new or worsening confusion, falls and nonspecific illness and less likely to present with fever and cough than nonfrail older adults. This is in keeping with other studies [14,32] and characterises the “typical atypical” way in which older patients, particularly those living with frailty, often present with acute illness.

New or worsening confusion was present in 36.9% of all patients on admission and 57.0% of frail patients experienced new or worsening confusion at some point during their admission episode, either formally diagnosed as delirium or noted as a symptom by their healthcare team. High prevalence of delirium associated with COVID-19 infection has been identified in other studies [28], yet local and national guidelines often fail to emphasise the importance of delirium as a potential indicator of COVID-19 illness [33]. Therefore, our findings on the prevalence of delirium and the range of other presenting features of COVID-19 in hospitalized older adults support low thresholds for COVID-19 testing in this population group [5] during periods of significant community viral transmission.

Overall, in-hospital mortality was 34.6%, which reflects the severity of the disease and the older age of our population and is similar to other studies [22,23]. Frail patients were more likely to die than nonfrail patients, and a dose–response relationship was observed, which persisted after adjustment for age, sex, illness acuity and multimorbidity. This finding supports measurement of frailty in this patient group to aid identification of older adults most vulnerable to this disease. Early identification of patients likely to have a complex clinical course allows for early advanced care planning, involving the patient and their families in treatment decisions, and patients can be prioritized for interventions known to improve outcomes in older hospitalized adults; for example, comprehensive geriatric assessment delivered by multidisciplinary teams on specialist medicine for the elderly wards [34]. However, frailty has not been consistently associated with mortality across all reports, with some supporting our findings [22,35], another equivocal [27] and one reporting no association [28]. Differing results to date may reflect small sample sizes and it is likely that large multicentre studies or meta-analyses will be required. Interestingly, results from the COPE study, which includes over 1500 patients with COVID-19 from several centres across the United Kingdom and Italy, support an association between higher frailty and higher mortality from COVID-19 [35].

There is considerable interest in understanding the pathogenesis of COVID-19 and pathways to mortality. We observed that 28.5% of our patient cohort had a normal CXR around the time of COVID-19 presentation and only 34.6% had classic or probable CXR features suggestive of the disease. This is similar to a study in 64 younger patients (mean age of 56 years) [36] and reinforces the message that a negative result does not exclude COVID-19 illness [30,37]. We also note that this observation was exaggerated in frail patients, with only 1 in 4 patients having classical or probable CXR changes on presentation compared to 1 in 2 older adults without frailty. This could reflect the underlying vulnerability of frail patients, who may be less able to compensate for the effects of COVID-19 illness and present before the radiological features of lung inflammation have developed. However, it could also indicate that the manifestations of the illness differ by level of frailty.

Other reports have suggested that there may be differences in the physiological response to COVID-19 infection by frailty status. For example, another cohort study found inflammatory responses blunted in frail compared to nonfrail patients [28]. This is consistent with our findings in relation to CRP and IL-6, which were significantly higher in nonfrail patients. It is possible that frail older adults are not able to mount strong immune responses to COVID-19 infection due to immunosenescence [38]. Higher inflammatory responses to COVID-19 have been associated with poorer outcomes [39], and dexamethasone, an immunosuppressant therapy, reduced mortality in patients with COVID-19 requiring oxygen therapy or mechanical ventilation [40]. Therefore, if immune responses differ between frail and nonfrail patients, this may have implications for the likely effectiveness of different treatments, and trials examining potential therapies should include measurement of frailty in older adults. It is also possible, as with the differences in CXR features, that frail patients are presenting earlier in their illness trajectory and hence have lower levels of inflammatory markers on presentation. However, we also found that higher proportions of frail compared to nonfrail older patients presented with raised HsTNI levels, and there was a nonsignificant trend for higher neutrophil count. These findings hint at potential alternative pathological consequences of COVID-19 infection in older adults with frailty, with cardiac complications and bacterial superinfection perhaps more likely eventual causes of death than high levels of systemic inflammation.

Our data did not confirm a difference in the LOS between frail and nonfrail patients, unlike other studies [35], although there was a trend in this direction and frail patients were more likely to stay in hospital beyond their clinically fit date. Delayed transfer of care has been previously observed in association with frailty and likely reflects difficulties in sourcing social care or onward care facilities, such as inpatient rehabilitation centres and care homes [41]. Consistent with this, frail patients had lower mobility on discharge than nonfrail patients.

Nonfrail patients were more likely to be admitted to the intensive care wards compared to frail patients (29% vs. 0.7%, *p* < 0.001). This suggests that in our centre, decisions regarding admissions to critical care were considered appropriately in our cohort in keeping with clinical guidelines [42]. It should be noted, however, that decisions on access to critical care or mechanical ventilation for older adults should remain individualized and take into consideration patients’ preference and goals of care [26]. Such decisions are not based on any one factor but a holistic patient assessment, weighing up the risks and benefits of intensive care treatment.

Our study has some limitations. It is a single-centre study, and our results are not generalisable to the whole population. For example, the older population in our local area is a highly homogenous demographic consisting mostly of individuals of White ethnicity. The impact of ethnicity on the morbidity and mortality of this disease has been widely reported [43,44] but we were not able to add to this. The sample size was also relatively small after accounting for inpatient mortality, thus limiting the potential to detect differences between frail and nonfrail patients with respect to secondary outcome measures. Additionally, only routinely collected data were available and some patients were not assessed by their treating team for frailty, necessitating scoring by the research team who may have been aware of the outcome in some cases. However, scoring was done by physicians experienced in frailty assessment at our centre and based only on information documented in the EHR on the patient’s admission to hospital, a validated method [31]. Finally, more detailed information regarding Intensive Care Unit (ICU) admission, treatments and mechanical ventilation was not included, as this is currently being evaluated in a larger study at our institution.

The absence of CFS in some of our patients, necessitating scoring by the research team, was partly due to the inclusion of patients ≥65 years old, whereas routine frailty scoring in our centre is only mandated for patients aged ≥75 years. Thus, the median age of those patients scored by the research team was younger than patients assigned scores by their treating team. However, it may also suggest the unfamiliarity of some clinicians with this scoring system, though widespread. As a result of changes to workforce organisation in preparation for the pandemic, many clinicians who were not familiar with modern geriatric medicine practices were incorporated into acute and general medicine rotas and older people’s therapy teams were temporarily redeployed. This may have contributed to lower-than-expected assessment of frailty by treating teams and supports the need for local educational and quality improvement work to increase awareness of frailty, its clinical assessment and association with patient outcomes [26,41].

The main strengths of our study are that it offers a detailed description of the clinical presentation, laboratory profile and inpatient trajectory of a cohort of hospitalised older adults with COVID-19 and includes a description of these patients by level of frailty. Data were retrieved digitally from the EHR or manually using a standardised data collection tool and missing data were limited. Also, the use of an EHR system in our hospital enabled clinicians to review patient medical records remotely, without the need for research staff to enter clinical areas and increase the risk of infection.

## 5. Conclusions

Older adult inpatients with COVID-19 infection are likely to present with atypical symptoms, develop delirium and have a high mortality, especially if they are also living with frailty. Clinicians should have a low threshold for testing for COVID-19 in older and frail patients when there is community transmission of COVID-19, and diagnosis should prompt early advanced care-planning discussions.

## Figures and Tables

**Figure 1 geriatrics-06-00011-f001:**
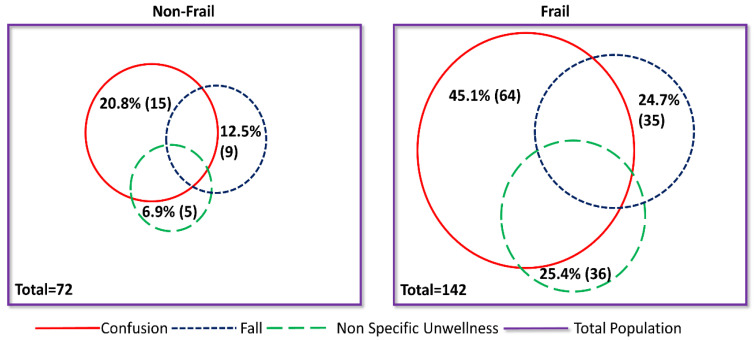
Venn Diagrams illustrating the prevalence of common geriatric illness by frailty with COVID-19 infection. Nonfrail: Clinical Frailty Score 1–4; Frail: Clinical Frailty Score 5–8; NOS: Nonspecific illness.

**Figure 2 geriatrics-06-00011-f002:**
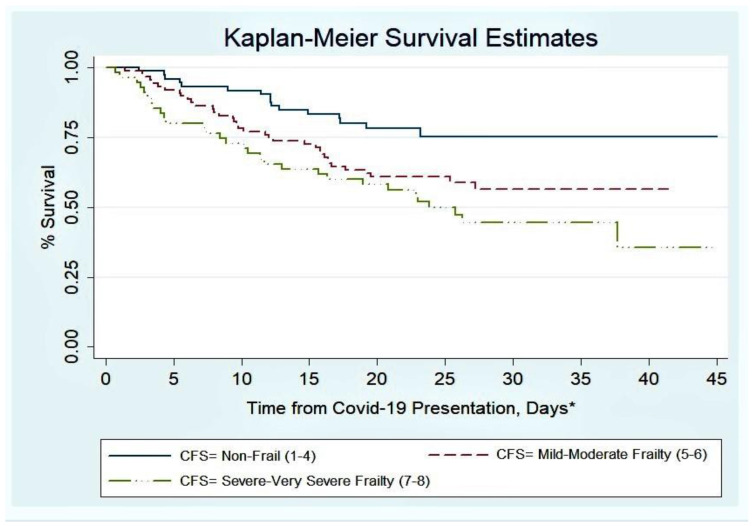
Survival after presentation with COVID-19 infection by Clinical Frailty Scale (CFS) Score. ***** The time from COVID-19 presentation is defined as the number of days from either admission to hospital or time from first positive swab if hospital acquired disease. For those discharged alive, follow-up was extended for the first 14 days after discharge.

**Table 1 geriatrics-06-00011-t001:** Characteristics of the whole cohort and by frailty (CFS 1–4 “Nonfrail” versus CFS 5–8 “Frail”).

Characteristic, Median (IQR)	All (*n* = 214)	Non-Frail (*n* = 72)	Frail (*n* = 142)	*p*
Age, years	80 (75, 87)	75 (70, 77)	84 (77, 89)	<0.001
Sex, % (*n*) women	43.9 (94)	29.2 (21)	51.4 (73)	0.002
CFS	6 (4, 7)	3 (2, 4)	6 (6, 7)	<0.001
BMI, Kg/m^2^	24.6 (21.2, 29.2)	26.5 (23.3, 31.2)	23.5 (20.1, 27.7)	0.0015
Known Dementia, % (*n*)	27.1 (58)	1.4 (1)	40.1 (57)	<0.001
Admission from care home, % (*n*)	31.3 (67)	0 (0)	47.3 (67)	<0.001
Pre-admission mobility, % (*n*)				<0.001
Independent	39.3 (84)	84.7 (61)	16.2 (23)	
Stick	15.0 (32)	9.7 (7)	17.6 (25)	
Frame	33.6 (72)	5.6 (4)	47.9 (68)	
Immobile	12.2 (26)	0 (0)	18.3 (26)	
Symptoms, % (*n*)				
Fever	73.4 (157)	81.9 (59)	69.0 (98)	0.043
Cough	63.6 (136)	76.4 (55)	57.0 (81)	0.005
SOB	59.8 (128)	62.5 (45)	58.5 (83)	0.568
Fatigue	48.1 (103)	62.5 (45)	40.8 (58)	0.003
Myalgia	14.5(31)	26.4 (19)	8.5 (12)	<0.001
Confusion	36.9 (79)	20.8 (15)	45.1 (64)	0.001
Fall	20.6 (44)	12.5 (9)	24.7 (35)	0.038
N & V	18.7 (40)	25.0 (18)	15.5 (22)	0.092
Diarrhoea	16.4 (35)	31.9 (23)	8.5 (12)	<0.001
Abdominal pain	7.0 (15)	12.5 (9)	4.2 (6)	0.025
Taste/ smell	4.7 (10)	9.7 (7)	2.1 (3)	0.013
Non-specifically unwell	19.2 (41)	6.9 (5)	25.4 (36)	0.001
Multimorbidity, % (*n*)				<0.001
<2 LTC	7.5 (16)	16.7 (12)	2.8 (4)	
2 LTC	9.8 (21)	16.7 (12)	6.3 (9)	
≥3 LTC	82.7 (177)	66.7 (48)	90.8 (129)	
Polypharmacy, % (*n*) 0–4 medications 5–9 medications ≥10 medications	21.0 (45) 51.9 (111) 27.1 (58)	36.1 (26) 45.8 (33) 18.1 (13)	13.4 (19) 54.9 (78) 31.7 (45)	<0.001
CXR on COVID-19 presentation % (*n*)				
Classical/Probable COVID-19	34.6 (74)	50.0 (36)	26.8 (38)	
Indeterminate for COVID-19	17.8 (38)	19.4 (14)	16.9 (24)	
Non COVID-19 Features	18.7 (40)	9.7 (7)	23.2 (33)	
Normal	28.5 (61)	20.8 (15)	32.3 (46)	0.002
Hospital Acquired Disease, % (*n*)	9.8 (21)	11.1 (8)	9.2 (13)	0.649
Time to first positive swab *, days	0 (0, 1)	0 (0, 1)	0 (0, 1)	0.290
Acuity, % (*n*) yes				
Low (NEWS2 <5)	45.3 (97)	50.0 (36)	43.0 (61)	
High (NEWS2 ≥5)	54.7 (117)	50.0 (36)	57.0 (81)	0.328
Vital signs, % (*n*)				
Fever >38.0 °C	28.5 (61)	40.3 (29)	22.5 (32)	0.007
Respiratory rate >24 bpm	38.8 (83)	36.1 (26)	40.1 (57)	0.568
Systolic blood pressure ≤90 mmHg	3.3 (7)	0 (0)	4.9 (7)	0.060
Pulse >130 bpm	5.1 (11)	5.6 (4)	4.9 (7)	0.845
Laboratory Values				
White blood cells, 10^9^/L	6.9 (5.0, 9.6)	6.3 (4.9, 8.6)	7.4 (5.1, 10.0)	0.094
Neutrophils, 10^9^/L	5.4 (3.7, 7.8)	4.8 (3.6, 7.2)	5.8 (3.8, 8.2)	0.19
Lymphocytes, 10^9^/L	0.67	0.61	0.67	0.28
	(0.48, 1.01)	(0.46, 0.90)	(0.50, 1.04)	
Creatinine, µmol/l	89.0	87.5	89.5	0.32
	(71.1, 118.5)	(69.5, 111.0)	(71.6, 128.6)	
C-Reactive Protein, % (*n*)				0.007
<40 mg/L	30.3 (65)	18.1 (13)	36.7 (52)	
40–100 mg/L	31.3 (67)	30.6 (22)	31.7 (45)	
>100 mg/L	37.4 (80)	50.0 (36)	31.0 (44)	
Interleukin-6, pg/mL	19.6 (7.3, 50.8)	26.8 (9.6, 71.1)	17.6 (6.6, 39.2)	0.04
Troponin, % (*n*) ≤58.1 ng/L >58.1 ng/L	50.9 (109) 23.8 (51)	68.1 (49) 16.7 (12)	42.3 (60) 27.5 (39)	0.009
D Dimer, % (*n*)				
≤230 ng/mL	15.6 (35)	16.7 (12)	16.2 (23)	
>230 ng/mL	54.2 (116)	54.1 (39)	54.2 (77)	0.942

* excluding 8 patients who were swabbed in the community and 1 patient who never had a positive swab; IQR: Interquartile range; CFS: Clinical Frailty Scale; Non-rail = CFS 1–4 and Frail = CFS 5–8; BMI: body mass index (available for 63 nonfrail and 130 frail patients); LTC: Long-term condition(s); CXR: Chest X-ray; Interleukin-6 (available for 42 non-rail and 85 frail patients). Percentages may not add up to 100% due to missing data.

**Table 2 geriatrics-06-00011-t002:** Hospital Trajectory and Outcome of the whole cohort and by frailty (CFS 1–4 “Nonfrail” versus CFS 5–8 “Frail”).

Hospital Trajectory and Outcome	All (*n* = 214)	Non-Frail (*n* = 72)	Frail (*n* = 142)	*p*
Ward moves, % (*n*)				0.021
Up to 1	42.1 (90)	44.4 (32)	40.8 (58)	
2	33.2 (71)	27.8 (20)	35.9 (51)	
3	15.4 (33)	12.5 (9)	16.9 (24)	
4 or more	9.4 (20)	14.3 (11)	6.4 (9)	
Admission to High Care, % (*n*)	10.3 (22)	29.2 (21)	0.7 (1)	<0.001
Delirium or new confusion during admission at any point, % (*n*)	47.7 (102)	29.2 (21)	57.0 (81)	<0.001
Mobility at discharge *, % (*n*)				<0.001
Independent	35.0 (49)	75.4 (43)	7.2 (6)	
Stick	5.7 (8)	3.5 (2)	7.2 (6)	
Frame	37.1 (52)	17.5 (10)	50.6 (42)	
Immobile	20.0 (28)	0 (0)	33.7 (28)	
Length of stay, days ^#^	11 (6, 18)	8 (4, 17)	12 (7, 19)	0.08
Hospital outcomes, % (*n*)				
Inpatient Death	34.6 (74)	20.8 (15)	41.6 (59)	0.003
Death as inpatient or within 14 days of discharge	38.3 (82)	22.2 (16)	46.7 (66)	0.001
Prolonged length of stay (>10 days)	53.5 (114)	45.1 (32)	57.8 (82)	0.08
New Institutionalisation *	5.6 (12)	3.5 (2)	12.1 (10)	0.08
Readmission *	19.3 (27)	15.8 (9)	21.7 (18)	0.385
Delayed transfer of care *	46.4 (65)	26.3 (15)	60.2 (50)	<0.001

^#^ values presented as median with interquartile range; * *n* = 140 (those who died excluded); CFS: Clinical Frailty Scale; Nonfrail = CFS 1–4 and Frail= CFS 5–8. Percentages may not add up to 100% due to missing data.

**Table 3 geriatrics-06-00011-t003:** Cox Proportional Hazards Regression Analysis: relationships between age, sex, frailty, illness acuity, multimorbidity and death.

	Univariable Analysis		Multivariable Analysis	
Characteristic	HR	95% Confidence Interval	HR	95% Confidence Interval
Age, years	1.04	1.02, 1.07	1.04	1.01, 1.07
Sex				
Women	Ref		Ref	
Men	1.47	0.94, 2.30	2.03	1.27, 3.24
CFS				
1–4	Ref		Ref	
5–6	1.99	1.11, 3.59	1.78	0.90, 3.53
7–8	2.83	1.54, 5.19	2.57	1.26, 5.24
Acuity, NEWS2				
Low (<5)	Ref		Ref	
High (≥5)	2.30	1.42, 3.65	2.33	1.45, 3.74
Multimorbidity				
0–1 LTC	Ref		Ref	
2 LTC	0.99	0.30, 3.28	0.89	0.22, 2.68
≥3 LTC	1.44	0.58, 3.58	0.85	0.32, 2.30

HR: Hazard Ratio; CFS: Clinical Frailty Scale; NEWS2: National Early Warning Score 2; LTC: long term condition(s). HRs adjusted for all co-variables in multivariable models.

## Data Availability

The data that support the findings of this study are available from the corresponding author, but restrictions apply to the availability of these data, which were used under license for the current study and so are not publicly available. Data are, however, available from the authors upon reasonable request subject to permission being obtained from Cambridge University Hospitals.

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
