# Peer review of "Clinical Features, Inpatient Trajectories and Frailty in Older Inpatients with COVID-19: A Retrospective Observational Study"

_geriatrics, 2021, doi:10.3390/geriatrics6010011_

Round 1

Reviewer 1 Report

The MS is an interesting article that describes the clinical features and inpatient trajectories of older 18 adults hospitalized with COVID-19, and explore relationships with frailty. The study is well designed. Conclusions and data analysis are properly made. 

Author Response

We thanks the reviewer for their comments.

Reviewer 2 Report

Introduction

Great summary of the literature. In the discussion, the authors comment on how this study adds to that existing literature but I would consider highlighting it in the introduction as well.

Methods

For those unfamiliar with the Clinical Frailty Score, it may be helpful to specify in the methods what it is and how it is assessed.

You note the citation but can you clarify in the text on the reproducibility of a post-hoc assessment of CFS by a researcher using the EHR? Given the relatively qualitative nature of the CFS, can you elaborate on how is this obtained by researchers (i.e. is it based on manual review of notes?).  Were the researchers blinded to outcomes when they reviewed the chart?

How are patients with a recorded CFS different from patients without a recorded CFS? This seems like a variable that may not be missing at random and further description of the patients in whom it was missing and how that relates to your outcomes may be helpful.

Results

I would consider separating Table 1 into 2 tables with one containing population characteristic s and the second containing your outcomes of interest. I got a bit lost on the length of table 1 and it might help to distinguish between these two.

I am not sure how Figure 1 adds to the manuscript. The numbers are the most helpful and they already appear in the table.

The authors demonstrate that frailty is associated with decreased transfer to ICU and decreased survival. They mention in the discussion that this may reflect triage decisions re: who to transfer to higher level of care. Can the authors further elaborate on the process for transfer to ICU at their hospital and how this might influence their analysis? Is it that frailty decreases survival or that frailty decreases ICU admission/intubation which leads to worse survival?

Minor comments

Given the small sample size, you may be better off reporting median and interquartile range rather than mean and standard deviation for all variables in Table 1. It is also a bit confusing to go back and forth between these measures of frequency.

Table 1: How are the pre-admission mobility levels defined? I am not familiar with the significance of these terms/levels.

Table 1: some of these numbers are unusual and I wonder if you mixed up the % and n? For example fever – non-frail and frail have decimal points for N – I assume these are supposed to be for %?

Figure 2: colors are a bit hard to distinguish.

Reviewer 3 Report

The article describe the clinical features and inpatient trajectories of older 18 adults hospitalized with COVID-19, and explore relationships with frailty. Is a retrospective observational study. Results are clearly presented and with a good statistical analysis.

Author Response

We thank the reviewer for their comments.